# Overcoming Distribution Mismatch in Quantizing Image Super-Resolution Networks

## Abstract

Quantization is a promising approach to reduce the high computational complexity of image super-resolution (SR) networks. However, compared to high-level tasks like image classification, low-bit quantization leads to severe accuracy loss in SR networks. This is because feature distributions of SR networks are significantly divergent for each channel or input image, and is thus difficult to determine a quantization range. Existing SR quantization works approach this distribution mismatch problem by dynamically adapting quantization ranges to the variant distributions during test time. However, such dynamic adaptation incurs additional computational costs that limit the benefits of quantization. Instead, we propose a new quantization-aware training framework that effectively **O**vercomes the **D**istribution **M**ismatch problem in SR networks without the need for dynamic adaptation. Intuitively, the mismatch can be reduced by directly regularizing the variance in features during training. However, we observe that variance regularization can collide with the reconstruction loss during training and adversely impact SR accuracy. Thus, we avoid the conflict between two losses by regularizing the variance only when the gradients of variance regularization are cooperative with that of reconstruction. Additionally, to further reduce the distribution mismatch, we introduce selective distribution offsets to layers with a significant mismatch, which selectively scales or shifts channel-wise features. Our algorithm effectively reduces the mismatch in distributions with minimal computational overhead.

## 1 Introduction

Image super-resolution (SR) is a core low-level vision task that aims to reconstruct the high-resolution (HR) images from their corresponding low-resolution (LR) counterparts. Recent advances in deep learning (Dong et al., 2015; Kim et al., 2016; Lim et al., 2017; Zhang et al., 2018a;b) have led to astonishing achievements in producing high-fidelity images. However, the remarkable performance relies on heavy network architectures with significant computational costs, which limits the practical viability, such as mobile deployment.

To mitigate the computational complexity of neural networks, quantization has emerged as a promising avenue. Network quantization has proven effective in reducing computation costs without much loss in accuracy, particularly in high-level vision tasks, such as image classification (Choi et al., 2018; Hou & Kwok, 2018; Zhou et al., 2016). Nonetheless, when it comes to quantizing SR networks to lower bit-widths, a substantial performance degradation (Ignatov et al., 2021) occurs, posing a persistent and challenging problem to be addressed.

Such degradation can be attributed to the significant variance present in the activation (feature) distributions of SR networks. The feature distribution of a layer exhibits substantial discrepancies across different channels and images, which makes it difficult to determine a single quantization range for a layer. Early approach on SR quantization (Li et al., 2020a) adopts quantization-aware training to learn better quantization ranges. However, as observed in Figure 1, despite careful selection, the quantization ranges fail to align with the diverse values within the channel and image dimension, which we refer to as *distribution mismatch*.

Recent approaches aim to address this challenge by incorporating dynamic adaptation methods to accommodate the varying distributions. For instance, Hong et al. (2022b) leverage distribution mean and variance to dynamically adjust quantization ranges for each channel and Zhong et al. (2022)

employ input-adaptive dynamic modules to determine quantization ranges on a per-image basis. However, such dynamic adaptation modules introduce significant computational overhead. While adapting the quantization function to each image during inference might handle the variable distributions, the overhead compromises the computational benefits of quantization.

In this study, we propose a novel quantization-aware training framework that addresses the distribution mismatch problem, by introducing a new loss term that regulates the variance in distributions. While direct regularization of distribution variance demonstrates potential in reducing quantization errors in each quantized feature, its relationship with the reconstruction loss is questionable. We observe that concurrently optimizing the network with variance regularization and reconstruction loss can disrupt the image reconstruction process, as shown in Figure 2. Therefore, we introduce a cooperative variance regularization strategy, where the variance is regulated only when it collaborates harmoniously with the reconstruction loss. To determine the cooperative behavior, we assess whether the sign values of the gradients from each loss are the same. Consequently, we can effectively update the SR network to optimize both quantization-friendliness and reconstruction accuracy.

To further reduce the distribution mismatch in SR networks, we introduce the concept of selective distribution offsets for features that exhibit severe mismatch. We first observe that the distribution mismatch problem is more critical in the channel dimension compared to the image dimension (Figure 1). Moreover, we find that the degree of channel-wise mismatch varies across different convolutional layers. As shown in Figure 3, certain layers exhibit a large mismatch between the distribution means, while others show a large mismatch between the distribution deviations. Intuitively, the mismatch in distribution mean can be reduced by applying channel-wise shifting of the distributions and that of the deviation be reduced by scaling. On this basis, we leverage additional offset parameters that selectively shift or scale the channel-wise distributions based on the specific mismatch aptitude of the layer. While these selectively-applied offsets effectively mitigate the distribution mismatch, they only incur negligible overhead, around $\times 30$ smaller storage size overhead or $\times 100$ fewer BitOPs compared to existing works with dynamic modules.

The contributions of our work include:

- We introduce the first quantization framework to address the distribution mismatch problem in SR networks without dynamic modules. Our framework updates the SR network to be quantization-friendly and accurate at the same time.

- We identify the distinct distribution mismatch among different layers and further reduce the distribution mismatch by shifting or scaling largely mismatching features.

- Compared to existing approaches on SR quantization, ours achieves state-of-the-art performance with similar or less computations.

## 2 RELATED WORKS

**Image super-resolution.** Convolutional neural network (CNN) based approaches (Ledig et al., 2017; Lim et al., 2017) have exhibited remarkable advancements in image super-resolution (SR) task, but at the cost of substantial computational resources. The massive computations of SR networks have led to a growing interest in developing lightweight SR architectures (Dong et al., 2014; Hui et al., 2019; 2018; Zhang et al., 2018a; Jo & Kim, 2021). Furthermore, various lightweight networks are investigated through neural architecture search (Chu et al., 2021; Kim et al., 2019; Li et al., 2020b; Song et al., 2020; Li et al., 2021), knowledge distillation (Hui et al., 2018; 2019; Zhang et al., 2021), and pruning (Oh et al., 2022). While these methods mostly focus on reducing the network depth or the number of channels, our focus in this work is to lower the precision of floating-point operations with network quantization.

**Network quantization.** By mapping 32-bit floating point values of input features and weights of convolutional layers to lower-bit values, network quantization provides a dramatic reduction in computational resources (Cai et al., 2017; Choi et al., 2018; Esser et al., 2020; Jung et al., 2019; Zhou et al., 2016; Zhuang et al., 2018). Recent works successfully quantize various networks with low bit-widths without much compromise in network accuracy (Cai & Vasconcelos, 2020; Dong et al., 2019; Habi et al., 2020; Jin et al., 2020; Lou et al., 2020; Wang et al., 2019; Yang & Jin,

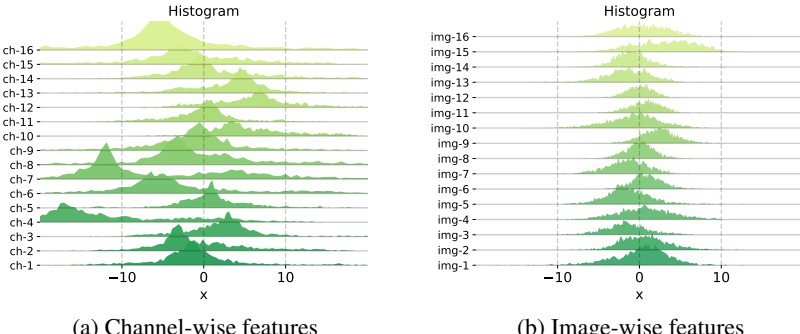

(a) Channel-wise features          (b) Image-wise features

Figure 1: **Distribution mismatch in SR networks.** SR networks exhibit a large mismatch inside the feature distributions, which results in a large quantization error. The mismatch is observed in both channel-dimension and image-dimension, but channel-wise mismatch is larger in magnitude and also more critical. Channels and images of a layer are randomly selected for visualization.

2021). However, these works primarily focus on high-level vision tasks, while networks for low-level vision tasks remain vulnerable to low-bit quantization.

**Quantized super-resolution networks.** In contrast to high-level vision tasks, super-resolution poses different challenges due to inherently high accuracy sensitivity to quantization (Ignatov et al., 2021; Ma et al., 2019; Xin et al., 2020; Wang et al., 2021). A few works have attempted to recover the accuracy by modifying the network architecture (Ayazoglu, 2021; Jiang et al., 2021; Xin et al., 2020) or by adopting different bits for each image (Hong et al., 2022a; Tian et al., 2023) or network stage (Liu et al., 2021). However, the key challenge of quantizing SR networks is in the vastly distinct feature distributions of SR networks. To deal with such issue, Li et al. (2020a) adopt a learnable quantization range for different layers. More recently, Hong et al. (2022b) recognize that the distributions are not only distinct per layer, but per channel and per input image and adopts dynamic quantization function for each channel. Moreover, Zhong et al. (2022) employ an input-adaptive dynamic module to adapt the quantization ranges differently for each input image. However, these dynamic adaptations of quantization functions during test-time cost non-negligible computational overheads. In contrast, instead of designing input-adaptive quantization modules, we focus on mitigating the feature variance itself. Our framework reduces the inherent distribution mismatch in SR networks with minimal overhead, accurately quantizing networks without dynamic modules.

# 3 PROPOSED METHOD

## 3.1 PRELIMINARIES

To reduce the heavy computations of convolutional layers in neural networks, the input feature (activation) and weight of each convolutional layer are quantized to low-bit values (Cai et al., 2017; Choi et al., 2018; Jung et al., 2019; Gholami et al., 2021). The input feature of the $i$-th convolutional layer $\boldsymbol{X}_i \in \mathbb{R}^{B \times C \times H \times W}$, where $B, C, H$, and $W$ denote the dimension of input batch, channel, height, and width, a quantization operator $Q(\cdot)$ quantizes the feature $\boldsymbol{X}_i$ with bit-width $b$:

$$Q(\boldsymbol{X}_i) = \text{Int}\left(\frac{\text{clip}(\boldsymbol{X}_i, \alpha_l, \alpha_u) - \alpha_l}{s}\right) \cdot s + \alpha_l, \tag{1}$$

where $\text{clip}(\cdot, \alpha_l, \alpha_u)$ truncates the input into the range of $[\alpha_l, \alpha_u]$ and $s = \frac{\alpha_u - \alpha_l}{2^b - 1}$. After truncation, the truncated feature is scaled to $[0, 2^b - 1]$, then rounded to integer values with $\text{Int}(\cdot)$, and it is rescaled to range $[\alpha_l, \alpha_u]$. To obtain better quantization ranges for SR networks, range parameters $\alpha_l, \alpha_u$ for each layer are generally learned through quantization-aware training (Li et al., 2020a; Zhong et al., 2022). Since the rounding function is not differentiable, a straight-through estimator (STE) (Bengio et al., 2013) is used to train the range parameters in an end-to-end manner. Following Zhong et al. (2022), we initialize $\alpha_u$ and $\alpha_l$ as the $j$-th and $100-j$-th percentile value of feature averaged among the training data. $j$ is set as 1 in our experiments to avoid outliers from corrupting

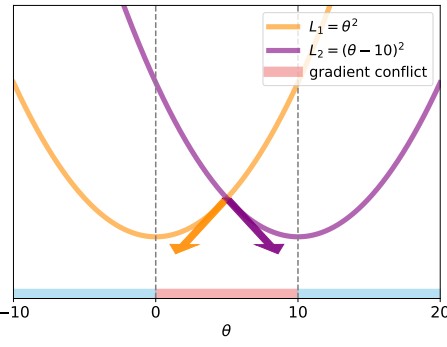 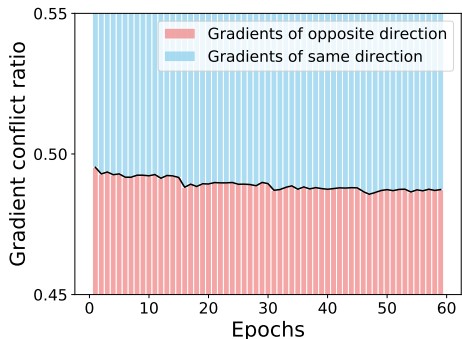

(a) Gradient conflict (Du et al., 2018)    (b) Gradient conflict ratio in EDSR

Figure 2: **Conflict between variance regularization and reconstruction loss.** Variance regularization updates a number of parameters in the *opposite* direction of reconstruction loss, which we refer to as gradient conflict. We plot the ratio of conflicted gradients during training when the two losses are jointly used. Nearly half of the parameters undergo gradient conflict which indicates that simply leveraging variance regularization and the reconstruction loss together can limit SR accuracy.

the quantization range. Similarly, to quantize the weight of the $i$-th convolutional layer $\boldsymbol{W}_i$, quantization operator $Q(\cdot)$ is used. However, instead of setting range parameters as learnable parameters, $\alpha_l, \alpha_u$ for weights are fixed as the $j$-th and $100-j$-th percentile of weights.

### 3.2 DISTRIBUTION MISMATCH IN SR NETWORKS

Quantization unfriendliness of SR networks is from the diverse feature (activation) distributions, as reported in previous studies (Li et al., 2020a; Hong et al., 2022b; Zhong et al., 2022), mainly due to the absence of batch normalization layers in SR networks. Existing SR quantization methods address this issue by employing one (Li et al., 2020a) or two (Zhong et al., 2022) learnable quantization range parameters for each convolutional layer feature. However, despite that the quantization-aware training process aims to find the optimal range for each feature, it fails to account for the channel-wise and input-wise variance in distributions. As illustrated in Figure 1, where notable discrepancies exist between layer-wise and channel-wise distributions, quantization grids are needlessly allocated to regions with minimal feature density. This mismatch in inter-channel distributions leads to performance degradation when quantizing SR networks. In the following sections, we introduce a new quantization-aware training scheme to address the distribution mismatch problem.

### 3.3 COOPERATIVE VARIANCE REGULARIZATION

Instead of focusing on finding a better quantization range parameter capable of accommodating the diverse feature distributions, our approach aims to regularize the distribution diversity beforehand. Obtaining an appropriate quantization range for a feature with low variance is an easier task compared to that of high variance. In this work, we define the overall mismatch of a feature distribution with the standard deviation,

$$M(\boldsymbol{X}_i) = \sigma(\boldsymbol{X}_i), \tag{2}$$

where $\sigma(\cdot)$ calculates the standard deviation of the feature. Thus, variance regularization can be directly applied to the feature to be quantized $(\boldsymbol{X}_i)$, which is formulated as follows:

$$\mathcal{L}_V(\boldsymbol{X}_i) = \lambda_V \cdot M(\boldsymbol{X}_i), \tag{3}$$

where $\lambda_V$ is the hyperparameter that denotes the weight of regularization. The overall $\mathcal{L}_V = \sum_i^{\#\ \text{layers}} \mathcal{L}_V(\boldsymbol{X}_i)$ is obtained by summing over all quantized convolutional layers. The variance regularization loss can be used in line with the reconstruction loss, which is originally used in the general quantization-aware training process. The optimization of parameter $\theta^t$ is formulated as:

$$\theta^{t+1} = \theta^t - \alpha^t(\nabla_\theta \mathcal{L}_R(\theta^t) + \nabla_\theta \mathcal{L}_V(\theta^t)), \tag{4}$$

where $\nabla_\theta \mathcal{L}_R(\theta^t)$ denotes the gradient from the original reconstruction loss and $\nabla_\theta \mathcal{L}_V(\theta^t)$ denotes the gradient from variance regularization loss and $\alpha^t$ denotes the learning rate. Updating the network to minimize the variance regularization loss will reduce the quantization error of each feature.

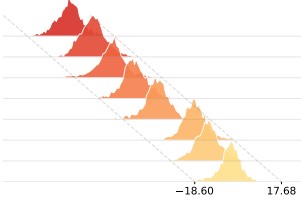 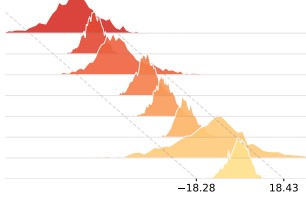 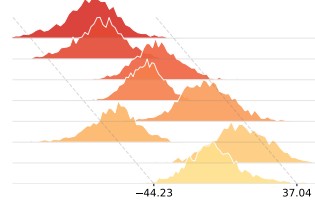

| (a) Small distribution mismatch | (b) Mismatch in channel-wise $\sigma$ | (c) Mismatch in channel-wise $\mu$ |
| --- | --- | --- |

Figure 3: **The distribution mismatch of different layers in EDSR.** While layer (a) shows an overall small mismatch between channels, layer (b) shows a large mismatch on deviation, and layer (c) exhibits a large mismatch on average. This motivates us to selectively *scale* features with large deviation mismatch ($m^{\sigma}$) and *shift* features with a large average mismatch ($m^{\mu}$).

However, then a question arises, *does reducing the quantization error of each feature lead to improved reconstruction accuracy?* The answer is, according to our observation in Figure 2, not necessarily. During the training process, the variance regularization loss can collide with the original reconstruction loss. That is, for some parameters, the sign of the gradient from reconstruction loss and that of variance regularization are opposing, referred to as gradient conflict (Du et al., 2018). As in Figure 2b, the ratio of parameters that undergo gradient conflict is not minor and such ratio persists throughout training, which means that regularization loss can hinder the reconstruction loss. However, we want to avoid the conflict between two losses, in other words, minimize the variance as long as it does not hinder the reconstruction loss. Thus, we determine whether the two losses are cooperative or not by examining the sign of the gradients of each loss. If the signs of the gradients are equal, then the parameter is updated in the same direction by two losses. By contrast, if the sign values are inverse, the two losses restrain each other, thus we only employ the reconstruction loss. In summary, we leverage variance regularization for parameters that the gradients have the same sign value as that from the reconstruction loss:

$$\theta^{t+1} = \begin{cases} \theta^t - \alpha^t(\nabla_\theta \mathcal{L}_R(\theta^t) + \nabla_\theta \mathcal{L}_V(\theta^t)), & \nabla_\theta \mathcal{L}_R(\theta^t) \cdot \nabla_\theta \mathcal{L}_V(\theta^t) \geq 0, \\ \theta^t - \alpha^t(\nabla_\theta \mathcal{L}_R(\theta^t)), & \nabla_\theta \mathcal{L}_R(\theta^t) \cdot \nabla_\theta \mathcal{L}_V(\theta^t) < 0. \end{cases} \quad (5)$$

This allows the network to reduce the quantization error cooperatively with the reconstruction error.

### 3.4 SELECTIVE DISTRIBUTION OFFSETS

The variance of the distribution can be reduced to a certain extent via variance regularization in Section 3.3. However, since regularization is applied only when it is cooperative with the SR reconstruction, the gap between distributions remains. In this section, we explore the remaining gap between distributions. First, as visualized in Figure 1, we observe that the distribution gap is larger (and more critical) in the channel dimension compared to the image dimension.

Also, we find that this extent of the channel-wise gap in the distribution is different for each layer of the SR network, as shown in Figure 3. Some layers (Figure 3b) exhibit a larger mismatch in the distribution deviation, while others (Figure 3c) show a larger mismatch in the distribution average. The quantization errors of the layer with a large mismatch in distribution mean can be decreased by shifting the channel-wise feature. Similarly, the mismatch in layers with large divergence in distribution deviation can be reduced by scaling each channel-wise distribution.

Since channel-wise shifting and scaling incur computational overhead and not all layers are in need of additional shifting and scaling (Figure 3a), we selectively apply offset scaling/shifting to layers that can maximally benefit from it. The standards for our selection are derived by feeding a patch of images to the 32-bit pre-trained network and calculating the mismatch in average/deviation of each layer. Given the $i$-th feature statistics $\tilde{X}_i$ from the pre-trained network, the mismatch of $i$-th convolutional layer is formulated as follows:

$$m_i^{\mu} = \sigma(\mu_c(\tilde{X}_i)) \quad \text{and} \quad m_i^{\sigma} = \sigma(\sigma_c(\tilde{X}_i)), \quad (6)$$

where $\mu_c(\cdot)$ and $\sigma_c(\cdot)$ respectively calculate the channel-wise mean and standard deviation of a feature and $\sigma(\cdot)$ calculates the standard deviation. After all $m_i^{\mu}$s and $m_i^{\sigma}$s ($i = 1, \cdots, \#\text{layers}$)

---

**Algorithm 1** Quantization-aware training process of ODM

---

**Input:** Pre-trained 32-bit network $\mathcal{P}$, distribution offset ratio $p$.
**Output:** Quantized network $\mathcal{Q}$.

   Using $\mathcal{P}$, obtain $m_i^\mu$ and $m_i^\sigma$ $(i = 1, \cdots, \#\text{layers})$ using Eq. 6
   **for** $i = 1, \cdots, \#$ layers **do**
      Initialize quantization range parameters $\alpha_l, \alpha_u$ for feature $\boldsymbol{X}_i$ of $\mathcal{Q}$ using percentile
      **if** $m_i^\mu$ is in top-$p$ ratio among $\boldsymbol{m}^\mu$ **then**
         Shift $\boldsymbol{X}_i$ with $\boldsymbol{s}^\mu$ using Eq. 7
      **if** $m_i^\sigma$ is in top-$p$ ratio among $\boldsymbol{m}^\sigma$ **then**
         Scale $\boldsymbol{X}_i$ with $\boldsymbol{s}^\sigma$ using Eq. 8
      Given $\boldsymbol{X}_i$, obtain variance regularization loss $\mathcal{L}_V(\boldsymbol{X}_i)$ using Eq. 3
      Given $\mathcal{L}_V$ and $\mathcal{L}_1$, update parameters of $\mathcal{Q}$ using Eq. 5

---

are collected, we apply additional scaling offsets to top-p layers with high $\boldsymbol{m}_i^\sigma$ value and shifting offsets to top-p layers with high $\boldsymbol{m}_i^\mu$ value. The shifting and scaling process for feature $\boldsymbol{X}_i$ of the $i$-th convolutional layer is formulated as follows:

$$\boldsymbol{X}_i^* = \boldsymbol{X}_i + \boldsymbol{s}^\mu, \quad \text{if } m_i^\mu \in \text{top-}p(\boldsymbol{m}^\mu), \tag{7}$$

$$\boldsymbol{X}_i^* = \boldsymbol{X}_i \cdot \boldsymbol{s}^\sigma, \quad \text{if } m_i^\sigma \in \text{top-}p(\boldsymbol{m}^\sigma), \tag{8}$$

where $\boldsymbol{s}^\mu, \boldsymbol{s}^\sigma \in \mathbb{R}^C$ are learnable parameters, top-$p(\cdot)$ constructs a set that contains values greater than the $100(1 - p)$-percentile value of the given set. $p$ is the hyperparameter that determines the ratio of layers to apply distribution offsets, which we set to $0.3$ in our experiments. Moreover, both offsets $\boldsymbol{s}^\mu$ and $\boldsymbol{s}^\sigma$ are quantized to low-bit, 4-bit in our experiments, to minimize the computational overhead. Consequently, the offsets additionally incur only $0.02\%$ overhead to the network storage size for EDSR. The offsets further relieve distribution mismatch with minimal overhead.

### 3.5 OVERALL TRAINING

Alg. 1 summarizes the overall pipeline for our framework, ODM. To update the parameters of the quantized network, including the selective offsets, we follow the common practice (Li et al., 2020a; Zhong et al., 2022) to use $\mathcal{L}_1$ and $\mathcal{L}_{\text{SKT}}$ for the reconstruction loss as follows:

$$\mathcal{L}_R = \mathcal{L}_1 + \lambda \mathcal{L}_{SKT}, \tag{9}$$

where $\mathcal{L}_1$ loss indicates the $l_1$ distance between the reconstructed image and the ground-truth HR image, and $\mathcal{L}_{\text{SKT}}$ loss is the $l_2$ distance between the structural features of the quantized network and the 32-bit pre-trained network. The structural features are obtained from the last layer of the high-level feature extractor. The balancing weight $\lambda$ is set as $1000$ in our experiments. Also, the weight $\lambda_V$ to balance $\mathcal{L}_V$ and $\mathcal{L}_R$ in Eq. 3 is set differently depending on the mismatch severeness of the SR architecture. We provide detailed settings in the supplementary materials.

## 4 EXPERIMENTS

The efficacy and adaptability of the proposed quantization framework ODM are assessed through its application to several SR networks. The experimental settings are outlined (Sec. 4.1), and both quantitative (Sec. 4.2) and qualitative (Sec. 4.3) evaluations are conducted on various SR networks. Ablation experiments are conducted to examine each component of the framework (Sec. 4.4).

### 4.1 IMPLEMENTATION DETAILS

**Models and training.** The proposed framework is applied directly to existing representative SR networks that produce satisfactory SR results but with heavy computations: EDSR (baseline) (Lim et al., 2017), RDN (Zhang et al., 2018b), and SRResNet (Ledig et al., 2017). Following existing works on SR quantization (Li et al., 2020a; Ma et al., 2019; Xin et al., 2020; Hong et al., 2022b; Zhong et al., 2022; Hong et al., 2022a), weights and activations of the high-level feature extraction module are quantized which is the most computationally-demanding. Training and validation are done with DIV2K (Agustsson & Timofte, 2017) dataset. ODM trains the network for 60 epochs,

| Model | Bit | Set5 | | Set14 | | B100 | | Urban100 | |
|---|---|---|---|---|---|---|---|---|---|
| | | PSNR | SSIM | PSNR | SSIM | PSNR | SSIM | PSNR | SSIM |
| EDSR | 32 | 32.10 | 0.894 | 28.58 | 0.781 | 27.56 | 0.736 | 26.04 | 0.785 |
| EDSR-PAMS | 4 | 31.59 | 0.885 | 28.20 | 0.773 | 27.32 | 0.728 | 25.32 | 0.762 |
| EDSR-DAQ | 4 | 31.85 | 0.887 | 28.38 | 0.776 | 27.42 | 0.732 | 25.73 | 0.772 |
| EDSR-DDTB | 4 | 31.85 | 0.889 | 28.39 | 0.777 | 27.44 | 0.732 | 25.69 | 0.774 |
| EDSR-ODM (Ours) | 4 | **32.03** | **0.891** | **28.48** | **0.779** | **27.49** | **0.735** | **25.79** | **0.778** |
| EDSR-PAMS | 3 | 27.25 | 0.780 | 25.24 | 0.673 | 25.38 | 0.644 | 22.76 | 0.641 |
| EDSR-DAQ | 3 | 31.66 | 0.884 | 28.19 | 0.771 | 27.28 | 0.728 | 25.40 | 0.762 |
| EDSR-DDTB | 3 | 31.52 | 0.883 | 28.18 | 0.771 | 27.30 | 0.727 | 25.33 | 0.761 |
| EDSR-ODM (Ours) | 3 | **31.80** | **0.888** | **28.35** | **0.776** | **27.41** | **0.732** | **25.52** | **0.770** |
| EDSR-PAMS | 2 | 29.51 | 0.835 | 26.79 | 0.734 | 26.45 | 0.696 | 23.72 | 0.688 |
| EDSR-DAQ | 2 | 31.01 | 0.871 | 27.89 | 0.762 | 27.09 | 0.719 | 24.88 | 0.740 |
| EDSR-DDTB | 2 | 30.97 | 0.876 | 27.87 | 0.764 | 27.09 | 0.719 | 24.82 | 0.742 |
| EDSR-ODM (Ours) | 2 | **31.49** | **0.883** | **28.12** | **0.770** | **27.26** | **0.727** | **25.15** | **0.756** |

Table 1: **Quantitative comparisons on EDSR** of scale 4.

| Model | Bit | Set5 | | Set14 | | B100 | | Urban100 | |
|---|---|---|---|---|---|---|---|---|---|
| | | PSNR | SSIM | PSNR | SSIM | PSNR | SSIM | PSNR | SSIM |
| RDN | 32 | 32.24 | 0.896 | 28.67 | 0.784 | 27.63 | 0.738 | 26.29 | 0.792 |
| RDN-PAMS | 4 | 30.44 | 0.862 | 27.54 | 0.753 | 26.87 | 0.710 | 24.52 | 0.726 |
| RDN-DAQ | 4 | 31.91 | 0.889 | 28.38 | 0.775 | 27.38 | 0.733 | 25.81 | 0.779 |
| RDN-DDTB | 4 | 31.97 | 0.891 | 28.49 | 0.780 | 27.49 | 0.735 | 25.90 | 0.783 |
| RDN-ODM (Ours) | 4 | **32.03** | **0.892** | **28.51** | **0.780** | **27.54** | **0.736** | **25.92** | **0.784** |
| RDN-PAMS | 3 | 29.54 | 0.838 | 26.82 | 0.734 | 26.47 | 0.696 | 23.83 | 0.692 |
| RDN-DAQ | 3 | 31.57 | 0.883 | 28.18 | 0.771 | 27.27 | 0.728 | 25.47 | 0.765 |
| RDN-DDTB | 3 | 31.49 | 0.883 | 28.17 | 0.772 | 27.30 | 0.728 | 25.35 | 0.764 |
| RDN-ODM (Ours) | 3 | **31.56** | **0.884** | **28.21** | **0.773** | **27.33** | **0.730** | **25.37** | **0.765** |
| RDN-PAMS | 2 | 29.73 | 0.843 | 26.96 | 0.739 | 26.57 | 0.700 | 23.87 | 0.696 |
| RDN-DAQ | 2 | 30.71 | 0.866 | 27.61 | 0.755 | 26.93 | 0.715 | 24.71 | 0.731 |
| RDN-DDTB | 2 | 30.57 | 0.867 | 27.56 | 0.757 | 26.91 | 0.714 | 24.50 | 0.728 |
| RDN-ODM (Ours) | 2 | **30.98** | **0.873** | **27.79** | **0.762** | **27.05** | **0.719** | **24.74** | **0.737** |

Table 2: **Quantitative comparisons on RDN** of scale 4.

with $1 \times 10^{-4}$ initial learning rate that is halved every 15 epochs and with a batch size of 8. All our experiments are implemented with PyTorch and run using a single RTX 2080Ti GPU.

**Evaluation.** We evaluate our framework on the standard benchmark (Set5 (Bevilacqua et al., 2012), Set14 (Ledig et al., 2017), BSD100 (Martin et al., 2001), and Urban100 (Huang et al., 2015)). We report peak signal-to-noise ratio (PSNR) and structural similarity index (SSIM (Wang et al., 2004)) to evaluate the SR performance. To evaluate the computational complexity of our framework, we measure the BitOPs and storage size. BitOPs is the number of operations that are weighted by the bit-widths of the two operands. Storage size is the number of stored parameters weighted by the precision of each parameter value.

### 4.2 QUANTITATIVE RESULTS

To evaluate the effectiveness of our proposed scheme, we compare the results with existing SR quantization works PAMS (Li et al., 2020a), DAQ (Hong et al., 2022b), and DDTB (Zhong et al., 2022) using the official code. To make a fair comparison with the existing works, we reproduce the results of other methods using the same training epochs. As shown in Table 1, our framework ODM outperforms other methods largely for all 4, 3, and 2-bit, and notably, the improvement is significant for 2-bit quantization. Also, 4-bit EDSR-ODM achieves closer accuracy to the 32-bit EDSR, where the margin is 0.07dB for Set5. This indicates that ODM can effectively bridge the gap between the quantized network and the floating-point network. Also, Table 2 compares the results on RDN. The results show that ODM achieves consistently superior performance on 4, 3, and 2-bit quantization.

Furthermore, we evaluate our framework on SRResNet which is shown in Table 3. SRResNet architecture includes BN layers and thus the distribution mismatch problem is not as severe as in EDSR

| Model | Bit | Set5 | | Set14 | | B100 | | Urban100 | |
|---|---|---|---|---|---|---|---|---|---|
| | | PSNR | SSIM | PSNR | SSIM | PSNR | SSIM | PSNR | SSIM |
| SRResNet | 32 | 32.07 | 0.893 | 28.50 | 0.780 | 27.52 | 0.735 | 25.86 | 0.779 |
| SRResNet-PAMS | 4 | 31.88 | 0.891 | 28.41 | 0.777 | 27.45 | 0.732 | 25.68 | 0.773 |
| SRResNet-DAQ | 4 | 31.85 | 0.889 | 28.41 | 0.777 | 27.45 | 0.732 | 25.70 | 0.772 |
| SRResNet-DDTB | 4 | 31.97 | 0.892 | 28.46 | 0.778 | 27.48 | 0.733 | 25.77 | 0.776 |
| SRResNet-ODM (Ours) | 4 | **32.00** | **0.892** | **28.46** | **0.778** | **27.48** | **0.734** | **25.77** | **0.776** |
| SRResNet-PAMS | 3 | 31.68 | 0.888 | 28.27 | 0.774 | 26.79 | 0.709 | 25.46 | 0.765 |
| SRResNet-DAQ | 3 | 31.81 | 0.889 | 28.35 | 0.776 | 27.40 | 0.733 | 25.63 | 0.772 |
| SRResNet-DDTB | 3 | 31.85 | 0.890 | 28.39 | 0.776 | 27.44 | 0.731 | 25.64 | 0.770 |
| SRResNet-ODM (Ours) | 3 | **31.86** | **0.890** | **28.39** | **0.776** | **27.44** | **0.732** | **25.65** | **0.771** |
| SRResNet-PAMS | 2 | 30.25 | 0.861 | 27.36 | 0.750 | 26.79 | 0.709 | 24.19 | 0.713 |
| SRResNet-DAQ | 2 | 31.57 | 0.886 | 28.19 | 0.773 | 27.30 | 0.729 | 25.39 | 0.765 |
| SRResNet-DDTB | 2 | 31.51 | 0.887 | 28.23 | 0.773 | 27.33 | 0.728 | 25.37 | 0.762 |
| SRResNet-ODM (Ours) | 2 | **31.59** | **0.887** | **28.27** | **0.773** | **27.36** | **0.729** | **25.44** | **0.765** |

Table 3: **Quantitative comparisons on SRResNet** of scale 4.

| Model | Bit | Storage size | BitOPs | PSNR | SSIM |
|---|---|---|---|---|---|
| EDSR | 32 | 1517.6K | 527.1T | 32.10 | 0.894 |
| EDSR-PAMS | 2 | 411.7K | 215.1T | 29.51 | 0.835 |
| EDSR-DAQ | 2 | 411.7K | 215.6T | 31.01 | 0.871 |
| EDSR-DDTB | 2 | 413.4K | 215.1T | 30.97 | 0.876 |
| EDSR-ODM (Ours) | 2 | **411.7K** | **215.1T** | **31.49** | **0.883** |
| RDN | 32 | 22271.1K | 6032.9T | 32.24 | 0.896 |
| RDN-PAMS | 2 | 1715.9K | 236.6T | 29.54 | 0.838 |
| RDN-DAQ | 2 | 1715.9K | 287.7T | 30.33 | 0.858 |
| RDN-DDTB | 2 | 1769.7K | 236.6T | 30.57 | 0.867 |
| RDN-ODM (Ours) | 2 | **1727.9K** | **236.6T** | **30.98** | **0.871** |
| SRResNet | 32 | 1546.8K | 588.8T | 32.10 | 0.894 |
| SRResNet-PAMS | 2 | 440.9K | 276.9T | 30.25 | 0.861 |
| SRResNet-DAQ | 2 | 440.9K | 279.0T | 31.57 | 0.886 |
| SRResNet-DDTB | 2 | 442.3K | 276.9T | 31.51 | 0.887 |
| SRResNet-ODM (Ours) | 2 | **441.4K** | **276.9T** | **31.59** | **0.887** |

Table 4: **Computational complexity comparison** with SR quantization methods on EDSR ($\times 4$).

or RDN. Nevertheless, ODM is also proven effective for quantizing SRResNet on all bit settings, showing slightly better performance than the existing quantization methods. Additional experiments that further demonstrate the applicability of ODM are provided in the supplementary materials.

Along with the SR accuracy, we also compare the computational complexity of our framework in Table 4. We calculate the BitOPs for generating a $1920 \times 1080$ image. Overall, our framework ODM achieves higher accuracy (PSNR/SSIM) with similar or less computational resources. As reported in Table 5, the distribution offsets incur minimal computational overhead. On EDSR, distribution offsets involve an additional $0.02\%$ storage size and the offset scaling and shifting involves $0.005\%$ additional BitOPs. In particular, RDN-ODM involves $\times 4$ smaller storage size overhead than RDN-DDTB, and $\times 800$ fewer BitOPs overhead than RDN-DAQ, while the PSNR gap is $0.41$dB or higher. Although the PSNR gap is smaller on SRResNet, ODM still achieves higher PSNR with fewer computations than DAQ and DDTB. Compared to existing works that utilize dynamic adaptation, the computational overhead is $\times 30$ smaller in storage size than DDTB and $\times 100$ smaller in BitOPs than DAQ. Compared to PAMS, our framework incurs additional storage size on RDN ($0.7\%$) and SRResNet ($0.1\%$), but the accuracy gap with PAMS is significant ($\sim 1.4$ dB).

### 4.3 QUALITATIVE RESULTS

Figure 4 provides qualitative results and comparisons with the output images from quantized EDSR and RDN. Our method, ODM, produces a further visually clean output image compared to existing quantization methods. In contrast, existing methods, especially PAMS, suffer from blurred lines

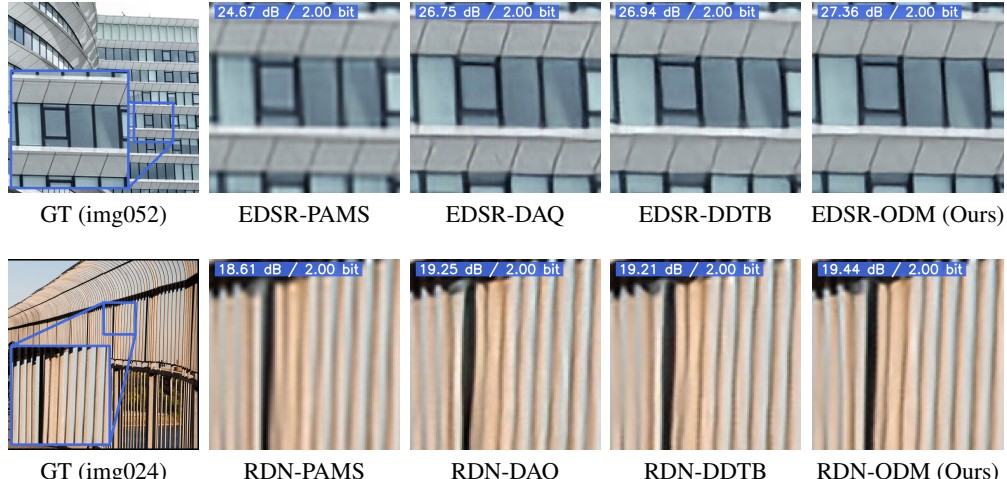

| GT (img052) | EDSR-PAMS | EDSR-DAQ | EDSR-DDTB | EDSR-ODM (Ours) |
| GT (img024) | RDN-PAMS | RDN-DAQ | RDN-DDTB | RDN-ODM (Ours) |

Figure 4: **Qualitative results** on Urban100 with EDSR and RDN-based models.

| Model | Coop. | Var. Reg. | Sel. Off. | Storage size | BitOPs | PSNR | SSIM |
|---|---|---|---|---|---|---|---|
| EDSR-PAMS | - | - | - | 411.7K | 215.0T | 29.51 | 0.835 |
| **(a)** | - | ✓ | - | - | - | 31.08 | 0.872 |
| **(b)** | ✓ | ✓ | - | - | - | 31.31 | 0.879 |
| **(c)** | - | - | ✓ | +0.08K (+0.02%) | +0.01T (+0.005%) | 31.40 | 0.880 |
| **EDSR-ODM** | ✓ | ✓ | ✓ | +0.08K (+0.02%) | +0.01T (+0.005%) | 31.49 | 0.883 |

Table 5: **Ablation study on each attribute of our framework.** Var. Reg. refers to the variance regularization loss and Coop. denotes whether the cooperative variance regularization is utilized or not, and Sel. Off. refers to the selective distribution offsets. Percentage in brackets denotes the additional computation compared to the baseline.

or artifacts. The qualitative results stress the importance of alleviating the distribution mismatch problem in SR networks. More results are provided in the supplementary materials.

### 4.4 ABLATION STUDY

In Table 5, we verify the importance of each attribute of our framework: cooperative variance regularization and distribution offsets. According to the results, cooperative variance regularization and distribution offsets respectively improve the baseline accuracy. Compared to using variance regularization directly (**a**), our cooperative scheme (**b**) improves the SR accuracy (+0.23dB). Although leveraging distribution offsets (**c**) incurs additional computations, it largely increases the accuracy while the computational overhead is minimal. Notably, when two components are jointly used, the accuracy increases compared to using each component separately, although the increase is relatively minor. This indicates that there is an overlapping effect of selective offsets and variance regularization on reducing the mismatch. Still, both components contribute to reducing the mismatch, resulting in a further accurate quantized SR network.

## 5 CONCLUSION

SR networks suffer accuracy loss from quantization due to the inherent distribution mismatch of the features. Instead of adapting resource-demanding dynamic modules to handle distinct distributions during test time, we introduce a new quantization-aware training technique that relieves the mismatch problem via distribution optimization. We leverage variance regularization loss that updates the SR network towards being quantization-friendly and also accurately super-resolving images. Also, through analysis of the distribution mismatch of different layers, we find that applying additional shifting offsets to layers with a large mismatch in terms of shift and scaling offsets to the layers with a large scaling mismatch can further reduce the distribution mismatch issue with minimal computational overhead. Experimental results demonstrate that the proposed training scheme achieves superior performance on various SR networks.

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
