# OpenReview forum: "Overcoming Distribution Mismatch in Quantizing Image Super-Resolution Networks"
_ICLR.cc/2024/Conference — Submitted to ICLR 2024_

### Official Review · Reviewer_v9HR · 2023-10-31

**Soundness:** 3 good
**Presentation:** 3 good
**Contribution:** 3 good
**Rating:** 6
**Confidence:** 4

**Summary:**

This paper explores how to overcome the distribution mismatch in quantizing SR. Specifically, the authors intuitively reduce the distribution mismatch by directly regularizing the variance of features when the gradients of variance regularization are cooperative with that of reconstruction. In addition, the authors introduce selective distribution offsets to layers with a significant mismatch, which selectively scales or shifts channel-wise features. The extensive experiments demonstrate the effectiveness of the proposed method.

**Strengths:**

1.	I enjoy the analyses of the distribution mismatch and conflict of the existing SR network. These observations are non-trivial and are critical for practical quantizing SR.
2.	The proposed method is novel and reasonable. The idea is simple yet effective. The authors have comprehensively demonstrated the proposed methods from the perspective of optimization.
3.	The paper is well-written and is easy to follow.

**Weaknesses:**

1.	It would be better if the authors can provide more details about Gradient conflict ratio.

2.	In addition to the variance regularization, the selective distribution offsets also employ a learnable parameter about the standard deviation of the features. It would be better if the author could provide more discussions about the relation of these two terms.

3.	Does the selective distribution offsets are learned on the features not processed in cooperative variance regularization?

**Questions:**

See the above weakness.

---

> ### Author Response · Authors · 2023-11-14
> **Response to Reviewer v9HR**
>
> We thank Reviewer v9HR for spending time reviewing and providing insightful feedback. We are encouraged by the positive comments on the novel method, the simple yet effective idea, and easy-to-follow writing. Below is the detailed response to each question.
>
>
> **Q1: More details about the gradient conflict ratio** \
> **A1:**
> Thanks for your helpful suggestion. Upon your advice, we updated the manuscript to add more details about the gradient conflict ratio.
> In the manuscript, we detailed how the gradient conflict ratio is measured: the ratio of parameters that the sign of gradient from reconstruction loss and that of regularization is different.
> Also, we additionally emphasized that, according to Figure 2 of the main paper, the conflict ratio decreases minimally during training, which indicates that variance regularization can hinder the reconstruction loss throughout training.
> Moreover, instead of disregarding the gradient of the regularization when the two gradient conflicts, we added analysis for weighting the regularization loss by the degree of conflict between two losses (measured by the cosine similarity).
> According to the results, it was slightly better to disregard the variance regularization loss when it is not cooperative.
>
> Table R3: **Analysis on the degree of gradient conflict** on EDSR x4 (2-bit). cos() measures cosine similarity.
> | Method    | Set5  | Set14 | B100  |Urban100|
> |:-------------|:-----:|:-----:|:-----:|:------:|
> | VR disregarded when $\nabla_\theta L_R \cdot \nabla_\theta L_V <0$ | 31.49 | 28.12 | 27.26 | 25.15  |
> | VR weighted with cos($\nabla_\theta L_R, \nabla_\theta L_V$)    | 31.45 | 28.12 | 27.25 | 25.14  |
>
>
> \
> **Q2: More discussion about the relation between variance regularization and selective distribution offsets** \
> **A2:**
> Thank you for your advice. As the reviewer mentioned, variance regularization and selective distribution offsets both function to reduce the distribution mismatch (feature-wise std).
> Each component largely reduces the mismatch and results in higher accuracy over the baseline.
> When the two components are jointly used, the accuracy increases compared to using each component separately, although the increasing amount is relatively minor.
> This indicates that there is, to some extent, an overlapping effect of selective offsets and variance regularization. Nevertheless, both components contribute to reducing the mismatch, resulting in a further accurate quantized SR network.
> We added the discussion to the manuscript.
>
>
>
> **Q3: Do the selective distribution offsets learned on the features not processed in cooperative variance regularization?** \
> **A3:**
> We would like to clarify that selective distribution offsets are also processed through cooperative variance regularization. The parameters of the quantized network (including the selective distribution offsets) are end-to-end trained. We updated the manuscript followingly.

---

> > ### Author Response · Authors · 2023-11-22
> > **Follow-up**
> >
> > Dear reviewer,
> >
> > We sincerely appreciate your constructive comments. We would like to follow up to check if our replies have effectively alleviated your concerns.
> > We are happy to answer further questions or concerns.
> >
> > Best,\
> > Authors

---

> > ### Comment · Reviewer_v9HR · 2023-11-23
> >
> > Thank you for your replies. I have no further questions and my concerns have been addressed.

---

### Official Review · Reviewer_p2GR · 2023-10-31

**Soundness:** 2 fair
**Presentation:** 2 fair
**Contribution:** 2 fair
**Rating:** 3
**Confidence:** 5

**Summary:**

This paper proposes a new quantization-aware training technique that relieves the mismatch problem via distribution optimization. Specifically, the authors use variance regularization loss, cooperative variance regularization and selective distribution offsets to reduce such mismatch. Experiments demonstrate the effectiveness of the proposed method.

**Strengths:**

This paper proposes a quantization framework to address the distribution mismatch problem in SR networks without dynamic modules. The proposed method achieves state-of-the-art performance with similar or less computations.

**Weaknesses:**

The experiment section can be improved. Please refer to the details below.

**Questions:**

1. The main motivation is that feature distributions of SR networks are significantly divergent for each channel or input image. In Figure 1, does the distribution mismatch only occur in the SR network? Does such a distribution mismatch occur in other networks? In addition, could you show the distribution after quantization?

2. In the experiments, the authors mainly use EDSR, EDN and SRResNet. However, these methods are very old. Could you compare the new SOTA SR networks, e.g., SwinIR?

3. In Table 1, for Bit=2, the results of EDSR-DAQ do not correspond to the original results of the DAQ paper. Could you discuss these results?

4. The experiments only address the scale of 4. It would be better to conduct more experiments on other scales and put the results in supplementary.

5. In the ablation study, could you conduct an experiment with only the cooperative variance regularization? In addition, the network with only the selective distribution offsets is comparable with Coop.+Var. Reg.+Sel. Off. This result demonstrates that Coop.+Var. Reg. are not  important.

---

> ### Author Response · Authors · 2023-11-14
> **Response to Reviewer p2GR**
>
> We thank reviewer p2GR for spending time reviewing and providing detailed feedback. We are encouraged that the reviewer recognized our state-of-the-art performance with similar or less computations. We address the reviewer's concerns and questions below in detail:
>
>
> **Q1-1: Does the distribution mismatch occur only in SR or other networks?** \
> **A1-1:**
> The distribution mismatch problem is especially severe for SR networks. For example, a classification network (ResNet20) shows a much more minor image-wise and channel-wise distribution mismatch compared to SR networks (EDSR, RDN). We measure the average variance of features on DIV2K validation set for SR networks and ImageNet validation set for the classification network. Thanks for your comments, we added such analyses to the supplementary material.
>
> Table R2: **Average feature mismatch**.
> | Model | Image-wise Variance  | Chanel-wise Variance |
> | ---------|:-----:|:-----:|
> | EDSR (x4) | 15.08 | 40.29 |
> | RDN (x4) | 6.40 | 58.14  |
> | ResNet-20 | 0.04 | 0.09  |
>
> \
> **Q1-2: The distribution after quantization** \
> **A1-2:**
> Thanks for your suggestion. Based on your comments, we added new figures to the supplementary material of distribution after quantization and the distribution after using our framework, ODM. The distribution of the ODM-applied network shows that ODM is effective in reducing the distribution mismatch.
>
>
> **Q2: Comparison with more recent models (e.g., SwinIR)** \
> **A2:**
> Please kindly refer to our supplementary material (Table S1), in which we reported results on more recent models SwinIR [A] and CARN [B]. According to the results, our method achieves consistent gain over existing methods also on recent models.
>
>
> **Q3: EDSR-DAQ does not correspond to the original results of the DAQ paper** \
> **A3:**
> We note that the result reported in the DAQ paper uses the EDSR backbone of 32 residual blocks (of 256 channel dimensions). In comparison, we use the EDSR-baseline backbone that consists of 16 residual blocks (of 64 channel dimensions), following PAMS and DDTB. We have reproduced EDSR-DAQ directly by the official codebase, which also matches the accuracy reported for EDSR-baseline in the official GitHub repository.
>
>
>
> **Q4: Experiments only on a scale of 4** \
> **A4:**
> Also, kindly refer to our supplementary material (Table S2) for the results on scale 2 SR models. The results show that our method is also beneficial for scale 2 SR models.
>
>
> **Q5-1: Ablation only using the cooperative variance regularization** \
> **A5-1:**
> If we understood correctly, Table 5 (b) presents the result of only using cooperative variance regularization.
>
>
>
>
> **Q5-2: Ablation indicates that the coop. var. reg is not important** \
> **A5-2:**
> We would like to emphasize that, compared to the baseline, using cooperative variance regularization (Coop. Var. Reg.) brings 0.8 dB gain in PSNR, which demonstrates that Coop. Var. Reg is important.
> The two main components we propose (Coop. Var. Reg. and Sel. Off.) both serve to reduce the distribution mismatch in SR; using two components together gives a relatively minor gain compared to the gain from each component (0.09 dB / 0.18 dB).
> However, this does not mean that each component is useless; each brings 0.8 dB / 0.9 dB gain over the baseline. It rather hints that the two attributes can have overlapping effects for reducing distribution mismatch.
>
> \
> [A] SwinIR: Image Restoration Using Swin Transformer, CVPR2021. \
> [B] Fast, Accurate, and Lightweight Super-Resolution with Cascading Residual Network, ECCV2018.

---

> > ### Author Response · Authors · 2023-11-22
> > **Follow-up**
> >
> > Dear reviewer,
> >
> > We would like to follow up to check if our replies have addressed your concerns. In the previous response, we made the following updates/clarifications:
> > * Regarding the experiment section, which was pointed out as the main weakness, we clarified in **A2** and **A4** that the suggested experiments were included in the supplementary material.
> > * Also, we clarified in **A3** that the reason for the difference in accuracy with the original DAQ paper is from using a different backbone. We follow DDTB and PAMS to use EDSR backbone of 16 residual blocks, while DAQ uses 32 blocks.
> > * Moreover, we integrated further analyses/clarifications into our manuscript, as detailed in **A1** and **A5**.
> >
> > We hope the replies have effectively alleviated your concerns.
> > We are happy to answer further questions or concerns.
> >
> > Best,\
> > Authors

---

### Official Review · Reviewer_Yy9i · 2023-11-09

**Soundness:** 3 good
**Presentation:** 3 good
**Contribution:** 3 good
**Rating:** 6
**Confidence:** 4

**Summary:**

The paper aims to deal with the inherent distribution mismatch of the features in quantizing image super-resolution. To this end, the paper introduces a variance regularization loss, which can cooperate well with the reconstruction loss by computing the signs of gradients. Furthermore, the paper proposes to apply shifting/scaling offsets to layers with a large mean/deviation. The proposed quantization framework ODM is evaluated on three representative SR models in the main paper. ODM exhibits better performance over competitors using a small storage size and low BitOPs.

**Strengths:**

The paper proposes the variance regularization loss, which can regularize the distribution diversity beforehand and cooperate well with the reconstruction loss. The selective distribution offsets further reduce the variance distribution. The proposed methods are based on analyses and observations. The experimental results are competitive by achieving high performance and reducing computation overhead. The writing is easy to follow.

**Weaknesses:**

The comparisons seem not fair in terms of training epochs, and the proposed method does not reduce BitOPs compared to the previous method, i.e., DDTB, which contradicts the motivation of the method.

**Questions:**

1. The authors reproduce the results of other methods using the same training epochs. Does the number of epochs influent the performance of other methods? Why not use their optimal training epochs for comparisons?
2. Compared to competitors, the authors use seemingly complicated methods to address the quantizing problem beforehand. Will the proposed method increase the training time?
3. We can observe from Tab. 4 that the proposed method achieves a better tradeoff between the storage size and BitOPs, to be precise. What makes the ODM need higher storage space than DAQ.
4. The verb is missing in the sentence after Eq.4.

---

> ### Author Response · Authors · 2023-11-14
> **Response to Reviewer Yy9i**
>
> We thank Reviewer Yy9i for the time spent reviewing and providing constructive feedback. We are encouraged by the positive comments on competitive performance and easy-to-follow writing. We address all raised comments and questions below in detail.
>
>
> **Q1-1: Fair comparison regarding training epochs** \
> **A1-1:**
> We follow DDTB [A] to use 60 epochs, and for a fair comparison, we reproduce PAMS [B] and DAQ [C] for 60 epochs using the official codebase.
>
> **Q1-2: Influence on the number of epochs** \
> **A1-2:**
> If we set the training epochs to 300 (the epochs reported in DAQ), the accuracies of the overall methods increase. However, we note that the order is preserved, and ours outperforms other methods.
>
> Table R1: **Different training epochs** on 2-bit EDSR for Urban100 PSNR.
> | Epochs | EDSR-PAMS  | EDSR-DAQ | EDSR-DDTB | EDSR-ODM (Ours) |
> | -------------------------------|:-----:|:-----:|:-----:|:------:|
> | 60   | 23.72 | 24.88 | 24.82 | 25.15  |
> | 300 | 24.09 | 24.90 | 25.01 | 25.51  |
>
> \
> **Q2: Comparison of training time** \
> **A2:**
> Using a single RTX 2080Ti GPU, the calibration for the selective offsets takes $\sim$30 seconds. The overall training time for ODM is $\sim$4.0 hours for quantizing EDSR, which is no less than that of DDTB ($\sim$4.0 hours).
>
>
> **Q3: What makes ODM need higher storage space than DAQ?** \
> **A3:**
> The additional storage space of ODM originates from the selective offset parameters. However, the storage overhead is relatively minimal compared to the significant bitOPs overhead of DAQ: as DAQ adaptively adjusts the quantization range parameters by calculating mean and variance at test-time, the bitOPs overhead is substantial.
>
> **Q4: Typo after Eq. (4)** \
> **A4:**
> Thank you for pointing out. We have revised our manuscript.
>
> **Q5: ODM does not reduce BitOPs compared to DDTB** \
> **A5:**
> We would like to note that, although the bitOPs of ODM is similar to that of DDTB, ODM occupies a smaller storage size, which is also an important computational cost benefit.
>
>
> \
> [A] Dynamic Dual Trainable Bounds for Ultra-Low Precision Super-Resolution Networks, ECCV 2022. \
> [B] PAMS: Quantized Super-Resolution via Parameterized Max Scale, ECCV 2020. \
> [C] DAQ: Channel-Wise Distribution-Aware Quantization for Deep Image Super-Resolution Networks, WACV2022.

---

> > ### Author Response · Authors · 2023-11-22
> > **Follow-up**
> >
> > Dear reviewer,
> >
> > We sincerely appreciate your insightful comments. We would like to follow up to check if our replies have effectively alleviated your concerns.
> > We are happy to answer further questions or concerns.
> >
> > Best,\
> > Authors

---

### Official Review · Reviewer_Bh7W · 2023-11-17

**Soundness:** 2 fair
**Presentation:** 2 fair
**Contribution:** 3 good
**Rating:** 6
**Confidence:** 3

**Summary:**

This manuscript focuses on quantizing super-resolution (SR) networks. Authors discover that the difficulty of quantizing SR networks is because of the fluctuation in activation distribution, which is significantly different in each channel. They propose ODM, a QAT framework to overcome the distribution mismatch problem by regularizing the variance in features using a new loss term. It mainly includes two contributions. First, it regularizes the gradients to ensure the losses are not in conflict. Second, it introduces a channel-wise offset that reduces the distribution mismatch.

**Strengths:**

The motivation is clear and strong. The authors design a new loss term and channel-wise quantization factors to regularize the activation to make the network easy to quantize. The plug-in module can be introduced and gain improvements in other networks and tasks that has variance feature distribution.

The experimental results are comprehensive. The proposed methods show consistent improvements on various SR networks and datasets (But the improvements are not that significant).

The figures and illustrations are easy to understand and targeted to the problem. And the overall writing is easy to follow.

**Weaknesses:**

There are lots of quantization methods that adopt channel-wise scaling and offsets. Although the channel-wise feature variance seems to be more severe in super-resolution networks, the channel-wise quantization factor is not novel.

The paper mainly solves one problem with two strategies. I wonder if they are repeated. The regularization loss makes the activation variance smaller in each channel, which is easy to quantize. And the channel-wise quantization factor quantizes the features in a channel-wise manner that will not be affected by the value differences between channels. The experimental results in the ablation study also show that the two methods are not orthogonal. Combining the two methods together can only outperform a little compared with solely using one of them.

The proposed methods may be too simple and need more insightful analysis and discoveries.

**Questions:**

x_i in Eq. (2) denotes the feature (activation). However, I wonder if it will lead to the homogenization of features since they are expected to have a low standard deviation. Did the authors try to minimize the difference in the mean of each channel?

---

> ### Author Response · Authors · 2023-11-20
> **Response to Reviewer Bh7W**
>
> We thank Reviewer Bh7W for contributing time reviewing and providing valuable feedback. We are encouraged that the reviewer found our work strongly motivated, easy to follow, and comprehensively experimented with. More importantly, we address the reviewer's concerns and questions below in detail:
>
>
> **Q1-1: Novelty in channel-wise quantization factor** \
> **A1-1:**
> As the reviewer also pointed out, there are works on SR quantization that adopt channel-wise quantization functions (e.g., DAQ [A]). However, we would like to note that simply adopting channel-wise scaling and offsets incur substantial computational overhead. Instead, our method adopts only scaling for particular layers or shifting for certain layers and even none for some layers, which effectively reduces the computation overhead (e.g., the bitOPs overhead is $\sim\times 100$ smaller compared to DAQ [A]). We find our *selective* approach for offsets to be helpful for achieving accurate quantization accuracy without incurring significant overhead.
>
> **Q1-2: Channel-wise quantization will not be affected by the value differences between channels** \
> **A1-2:**
> We would like to clarify that we utilize layer-wise quantization factor for each layer. For a few layers, we shift or scale the input of the convolutional layer with channel-wise offsets, which are then quantized with a layer-wise quantization factor. Thus, the quantization accuracy is affected by value differences in channels.
>
>
> **Q2: Repeated strategies for one problem** \
> **A2:**
> As the reviewer mentioned, both variance regularization and selective offset strategy alleviate the distribution mismatch problem in SR. Using two components together gives a relatively minor gain (0.09 dB / 0.18 dB) compared to the gain from each component (0.8 dB / 0.9 dB). This indicates that there is, to some extent, an overlapping effect of the two strategies. Nevertheless, both components reduce the mismatch, resulting in a further accurate quantized SR network. We will add further discussion of the overlap to our supplementary material.
>
>
>
> **Q3: More insightful analysis and discoveries** \
> **A3:**
> Thanks for your suggestion, we added deeper analyses on the distribution mismatch problem in Section S5.1 of the supplementary material: how solving such a problem is especially crucial for SR networks and how our method effectively reduces the mismatch. Also, as suggested by another reviewer, we added analysis on the gradient conflict problem in Section S5.2. Please kindly refer to the supplementary material for more details.
>
>
> **Q4: Will variance regularization lead to homogenization of features?** \
> **A4:**
> Directly using variance regularization can lead to the homogenization of features, which can trigger a PSNR loss, as shown in our ablation study. Since we use variance regularization loss cooperatively with the reconstruction loss, the features are not fully homogenized, as an example is visualized in Figure S2 of the supplementary material.
> Moreover, we tried regularizing the difference in the mean of each channel, which gave a slight increase in accuracy for a few settings. Thank you for your constructive suggestions, we will integrate such an analysis into our manuscript.
>
> Table R4: **Variance regularization** on EDSR x4 (2-bit).
> | Method    | Set5  | Set14 | B100  | Urban100 |
> |:-------------|:-----:|:-----:|:-----:|:------:|
> | Regularize std of feature | 31.49 | 28.12 | 27.26 | 25.15 |
> | Regularize std of ch-wise mean | 31.51 | 28.15 | 27.29 | 25.21 |
>
>
>
> \
> [A] DAQ: Channel-Wise Distribution-Aware Quantization for Deep Image Super-Resolution Networks, WACV2022.

---

> > ### Author Response · Authors · 2023-11-22
> > **Follow-up**
> >
> > Dear reviewer,
> >
> > We sincerely appreciate your inspiring comments. We would like to follow up to check if our replies have effectively alleviated your concerns.
> > We are happy to answer further questions or concerns.
> >
> > Best,\
> > Authors

---

### Author Response · Authors · 2023-11-14
**General Response to All Reviewers**

We thank all reviewers for their time and effort. We are encouraged that the reviewers found our work to have novel and reasonable methods (v9HR) that are based on observations (Yg9i, v9HR), to achieve competitive performance (Yg9i, p2GR), and also easy-to-follow (Yg9i, v9HR). More importantly, we address all comments one by one, but beforehand, we would like to make a general summary of our response. Also, we updated the paper accordingly, and the revised text is marked in blue for ease of review. We are happy to address further questions or concerns.

* **Fairness in comparison:**
We would like to clarify that we set the same training epochs for all methods (60 epochs) in which all results are reproduced by the official codebase, including DAQ [A]. The gain of our method over other methods is preserved when all methods are trained for more epochs (300 epochs). Please refer to Table R1 for detailed results.

* **Additional experiments:**
We would like to note that experiments on scale 2 SR models or more recent models (e.g., SwinIR [B]) are already provided in the supplementary material. The results show that our framework achieves consistent gain over existing methods on various SR models. Please kindly refer to the supplementary material for detailed results.

* **Additional analyses:**
We updated our manuscript based on constructive feedback to include additional analyses. For example, we added detailed analyses on the distribution mismatch to demonstrate that SR networks face more severe mismatch than classification networks and that applying our method effectively reduces the mismatch. Also, we added details on the gradient conflict and the relation between our two methods to reduce the distribution mismatch.

\
[A] DAQ: Channel-Wise Distribution-Aware Quantization for Deep Image Super-Resolution Networks, WACV2022. \
[B] SwinIR: Image Restoration Using Swin Transformer. CVPR2021.

---

### Meta-Review · Area_Chair_NT2s · 2023-12-13

**Metareview:**

This paper aims to address the issue of inherent distribution mismatch in the features of quantitative image super-resolution. The authors use variance regularization loss, collaborative variance regularization, and selective distribution shift to reduce this mismatch. Experiments have demonstrated the effectiveness of this method. The proposed approach shows very competitive results in terms of performance and computational efficiency. The main concern is the lack of rigour in the proposed method, and some aspects of the experimental setup are still not fully perfected.

**Justification For Why Not Higher Score:**

For this paper, the reviewers have provided completely opposing conclusions. The main concern of the reviewers is that the two strategies proposed by the authors are repetitive in purpose. The authors addressed this in their rebuttal with further discussion, but still did not fully resolve the issue. Additionally, improvements are needed in the experimental section to make the results more rigorous.

**Justification For Why Not Lower Score:**

N/A

---

### Decision · Program_Chairs · 2024-01-16

Reject